# Refining Preference-Based Recommendation with Associative Rules and Process Mining Using Correlation Distance

Mohd Anuaruddin Bin Ahmadon [1,*,†] , Shingo Yamaguchi [1,†] , Abd Kadir Mahamad [2] and Sharifah Saon [2]

1. Graduate School of Sciences and Technology for Innovation, Yamaguchi University, Tokiwadai 2-16-1, Ube 755-8611, Japan
2. Faculty of Electrical and Electronic Engineering, Universiti Tun Hussein Onn Malaysia, Batu Pahat 86400, Johor, Malaysia
* Correspondence: anuar@yamaguchi-u.ac.jp
† This paper is an extended version of our paper published in Mohd Anuaruddin Bin Ahmadon, Shingo Yamaguchi, "Mining Consumer Services Based on User Preference with Associative and Process Mining", Proc. of ZINC 2021, May 2021; pp. 185–190 (Received Special Merit Award).

**Abstract:** Online services, ambient services, and recommendation systems take user preferences into data processing so that the services can be tailored to the customer's preferences. Associative rules have been used to capture combinations of frequently preferred items. However, for some item sets X and Y, only the frequency of occurrences is taken into consideration, and most of the rules have weak correlations between item sets. In this paper, we proposed a method to extract associative rules with a high correlation between multivariate attributes based on intuitive preference settings, process mining, and correlation distance. The main contribution of this paper is the intuitive preference that is optimized to extract newly discovered preferences, i.e., implicit preferences. As a result, the rules output from the methods has around 70% of improvement in correlation value even if customers do not specify their preference at all.

**Keywords:** process mining; associative mining; personalization; recommendation; decision support

## 1. Introduction

In the context of online services, ambient services, and recommendation systems, user preferences are usually related to multiple attributes specified by the user based on certain factors such as environment, culture, and psychological factors. In general, preference analysis includes the perception and sentiment of the users toward selecting the target services or items. Users are becoming more attracted to tailored preferences in on-demand online services related to health, music, movies, food, and fashion. Online service providers are always keeping up with recommendation technology to tune and match user preferences so that current users will continue using their services. For some preference analyses, it is hard to recommend accurate results that match the user's experience due to the limitation of references in the database. A system that supports user decision-making or provides personalized services can only be implemented if the users explicitly define their preferences. In general, these systems can only provide excellent and meaningful results during certain events when these explicit and implicit preferences or actions take place. Based on the insight and related information, the systems can analyze the causal relationship between these references.

For recommendation of preferences, explicit and implicit references are always related to spatio-temporal concepts where the time and space of the users are considered. For example, a system can judge the user's motives for entering a space, such as entering a kitchen in a house to prepare breakfast in the morning. Smart systems can identify their implicit needs to suggest meals with low calories, such as coffee and scrambled eggs, by relating the reference to information on available ingredients in the house. The

information here shows that users may have many other choices for breakfast, but it is hard to decide what to choose. Spatio-temporal reasoning is usually effective when the references include a sequence of actions, frequencies, and repetitions of the user's behavior. However, recommending implicit preferences requires a method to analyze the causal relationship with the user's explicit preference and determine the best selection based on the relationship.

Learning and identifying the implicit preferences of users is challenging due to limited spatial-temporal references. Moreover, the causal relationship between explicit and implicit preferences must be strong enough for the recommendation to be meaningful and reliable. It is also essential to differentiate the user's *'needs'* and *'wants'*. Usually, the 'needs' of the users are defined explicitly. In contrast, identifying the 'wants' is the problem that should be addressed. Decisions are even harder for spatio-temporal needs to learn the specific attributes of the available options. Moreover, users are always influenced by the gain of making certain decisions and the fear of loss. Users usually do not make purely analytical decisions and are affected by sentiments, cultures, and psychological factors. Hence, we often see that users always make decisions by relying on other users' sentiments, news, or rumors. This is even harder for inexperienced or first-time users who spend extra time and effort comparing and choosing based on limited information.

The motivation of this research is many research focuses on finding explicit preferences from existing data such as product specification, sales log, reviews, and surveys but do not focus on highly related implicit preference. Explicit preference can be easily extracted from existing data mining methods such as clustering analysis, machine learning, associative mining, and genetic algorithm. However, the gap lies when extracting implicit preference [1]. He et al. [2] stated the problem in extracting implicit reference where even though the explicit preference was given, we need to determine the co-occurrences in the explicit preference of another user. However, there are some cases where users do not have any explicit preference at all. Implicit preference in our study is regarded as an *"unknown preference"* by the user. Moreover, the implicit preference must have strong relations with each other. It means that the customer only knows their implicit preference if some relation related to their explicit preferences is given. In this method, we use a combination of process and associative mining to extract implicit preferences even if the users do not specify their explicit preferences.

In this paper, we proposed a method to extract associative rules with a high correlation between X and Y based using process mining, associative mining, and correlation distance. The overview is shown in Figure 1. The figure shows that for first-time users, it is easier to make decisions intuitively. Therefore, they need an 'intuitive recommendation' based on their preference. Our approach takes a service log and extracts the preference model using process mining. Then the service model is refined by pruning associative rules. A result is an option for selection that discovers the implicit preference that even the user does not know before.

The main contributions of this paper are as follows:

1. A method that combines process mining and data mining to extract preference models for implicit preferences.
2. Extract implicit preferences that have a strong correlation between attributes even if the user sets some of their preferences as 'no-interest.'
3. The method outputs not a single choice of an item but multiple combinations of highly correlated items as recommendations to both first-time users and experienced users.

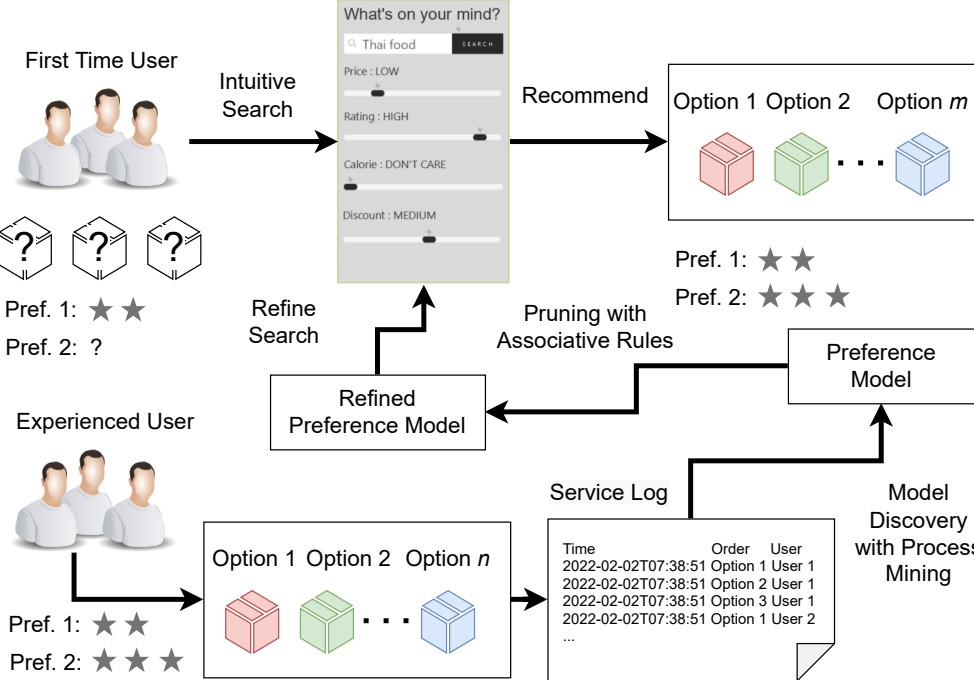

**Figure 1.** An overview of our proposed method.

The benefit of this study is that even though the users do not give any explicit preference, which is *'no preference'* set for all attributes of any item, the method can still extract what they might prefer by referring to the previous preference of previous customer as a starting point. If any explicit preferences are given, the preferences will be used as a reference. Intuitively, the user requires less effort but more flexibility in making a decision.

This paper is organized as follows; After the introduction, we give the preliminary to introduce process mining, association rule, and Cook's distance. Next, we define the problem of service preference extraction and explain the problem. Finally, we evaluate the proposed method to show its effectiveness.

## 2. Preliminary

### 2.1. Process Mining

Process mining [3] is a technique to extract process models from event logs such as sales logs, system logs, and business process logs. For example, a sales record of a transaction between a client and business or customer purchase records. We can utilize process mining to extract a process model called Petri net from an event log. Petri net [4] can represent the execution order or sequence of actions. Process mining links data and processes where some data can only be visualized as a process or as the action of a sequence. Process mining can be performed with process mining tools such as ProM [5], Disco [6], or RapidProM [7].

In this paper, we utilize a process mining method called inductive miner [8]. Inductive miner is a method to extract logically correct and sound processes. A logically correct process does not contain any conflict, such as deadlock or overflow. An inductive miner can extract sound Petri net or an equivalent form of Petri net called process tree [9]. The representation of the process using sound Petri net and process trees ensures that the process does not fall into spaghetti conditions. Therefore, the inductive miner is suitable for our approach. We utilize the Petri net to represent items' selections in sequences with strong correlation and frequency.

Concretely, the inductive miner extracts the process model in a block model representing a sequence, exclusive choice, parallel, and loop constructs. These four constructs represent the basic constructs in ost business process workflows. Business workflows, including customer services, can be modeled with Petri net.

### 2.2. Association Rule

An association rule represents a rule of antecedent and consequent. An association rule can be expressed with $\mathcal{A} \Rightarrow \mathcal{C}$ where $\mathcal{A}$ is the antecedent for consequent $\mathcal{C}$ such that when $\mathcal{A}$ occurs then $\mathcal{C}$ will occur. As an example, we take an association rule such as *outlook=sunny, windy=no* $\Rightarrow$ *play=yes* represents that if the outlook is sunny or windy, then do play outside. Association rule can be extracted from popular algorithms such as Apriori [10,11] or FPGrowth [12]. The rule is decided by proportional values such as support, confidence, and lift. To extract associative rules, we utilize the Apriori algorithm.

Let $\mathcal{A}$ or $\mathcal{C}$ be an itemset. Support $supp(L)$ is the number of instances that satisfy the rule. The confidence ratio is the ratio of a rule to be true in a dataset. It is given as in Equation (2).

$$conf(L \Rightarrow R) = \frac{supp(L \cup R)}{supp(L)} \tag{1}$$

Lift is the ratio of confidence and unconditional probability of the consequent. lift $> 1.0$ shows that $L$ and $R$ are dependent on each other.

$$lift(L \Rightarrow R) = \frac{supp(L \cup R)}{supp(L) \times supp(R)} \tag{2}$$

### 2.3. Cook's Distance

Cook's distance $Dist(i)$ [13] is used to measure the influence of an independent variable against another multi-dependent variable. The approach is by removing the target independent variable from the observation. The Cook's distance can be calculated with Equation (3). Equation (3) shows the average of $y$ at observation $j$ when $i$ is removed from the observation. $r$ is the regression model's coefficient.

$$Dist(i) = \frac{\sum_{j=1}^{n}(\hat{y}_{j(i)} - \hat{y}_j)^2}{r\hat{\sigma}^2} \tag{3}$$

The sum of $(\hat{y}_{j(i)} - \hat{y}_j)^2$ is calculated at each observation $j$ where the distance is calculated based on the regression line of each observation when $i$-th observation is removed. Since all points on the distribution are considered, Cook's distance is calculated based on the regression by the concept of 'leave-one-out.' We can say that Cook's distance is the distance between the point of regression line produced by averaged $y$ value and when $i$ is removed from the observation. The calculated distance can measure the influence of $i$ in the distribution group because Equation (3) averages the sum of residuals $y$ with the MSE.

The given Cook distance shown in Equation (3) utilizes Manhattan distance [14] when calculating the absolute sum of $\hat{y}_{j(i)} - \hat{y}_j$. It is the $L_1$-norm (Manhattan's generalized form) derived from a multidimensional numerical distance called Minkowski distance [15] as shown in Equation (4). Given an $L_p$-norm in Equation (4), the sum of the absolute value of $\hat{y}_{j(i)} - \hat{y}_j$ in Equation (3) satisfies $L_p$-norm when p $=$ 1.

$$L_p(\hat{y}_{j(i)}, \hat{y}_j) = \left(\sum_{j=1}^{n} |\hat{y}_{j(i)} - \hat{y}_j)|^p\right)^{\frac{1}{p}} \tag{4}$$

Cook's distance is simply the distance between averaged regression value of $\hat{y}_{j(i)}$ (when $i$ is removed) and the normal average value $\hat{y}_j$. By replacing the numerator with $L_2$-norm, we can obtain the Euclidean version of Equation (3).

Distance $Dist(i)$ is a metric if it satisfies (i) $\delta(x, y) \geq 0$ (non-negativity); (ii) $\delta(x, y) = \delta(y, x)$ (symmetry); (iii) $\delta(x, y) \geq 0$ if and only if $x = y$ (coincidence axiom); and (iv) $\delta(x, y) \leq \delta(x, z) + \delta(z, y)$ (triangular inequality axiom) where $\delta(x, y)$ is the distance between $x$ and $y$, and $z$ is the point between them. The properties (i), (ii), and (iii) are trivial when $x$ and $y$ are obtained from absolute value, and if $x$ and $y$ are the same, the distance is 0, also symmetrically, the distance is the same. To show Cook's distance satisfies the triangular inequality axiom in (iv), we can utilize the generalized form of Manhattan

distance and Euclidean distance shown in Equation (4). Since Cook's distance is $L_1$-norm, it is well known that we can set $p = 2$ to obtain the $L_2$-norm (Euclidean distance). Therefore, we can write the distance $Dist_{L_2}(i)$ shown in Equation (5). From Equation (5), we can identify that $Dist_{L_2}(i)$ is Euclidean and satisfies the triangular inequality axiom.

$$Dist_{L_2}(i) = \frac{1}{r\sigma^2} \sqrt{\sum_{j=1}^{n}(y_{j(i)} - \hat{y}_j)^2} \tag{5}$$

As stated by Chai et al. [16], the value of $\sigma^2$ satisfies triangular inequality (see Equation (6)). The regression coefficients are represented by $p$, and $\sigma^2$ is the regression's Mean Squared Error (MSE). The value of $\sigma^2$ is the Euclidean distance of residual error on how close a point $y$ is to the regression line (when averaging the $y$ values), assuming that the distribution is a Gaussian distribution. The value of $\sigma^2$ can be calculated as in Equation (6). The Euclidean distance is divided with $\sqrt{n}$ where $n$ is the number of samples in the distribution.

$$\sigma^2 = \sqrt{\frac{\sum_{j=1}^{n}(y_{(j)} - \hat{y}_j)^2}{n}} \tag{6}$$

The residuals $y_j(j = 1, 2, \cdots, n)$ can be represented by $n$-dimensional vector $V$. Let $X$, $Y$, and $Z$ be $n$-dimensional vectors. Based on the metric properties [17], Equation (7) shows that $\sigma^2$ satisfies the triangular inequality axiom [16] such that

$$\sqrt{\frac{1}{n}\sum_{j=1}^{n}(x_j - y_j)^2} \leq \sqrt{\frac{1}{n}\sum_{j=1}^{n}(x_j - z_j)^2} + \sqrt{\frac{1}{n}\sum_{j=1}^{n}(y_j - z_j)^2} \tag{7}$$

We can also obtain $L_2$-norm (the Euclidean version) of Cook's distance as follows:

$$Dist(i) = \frac{\sqrt{\sum_{j=1}^{n}(y_{j(i)} - \hat{y}_j)^2}}{r\sqrt{\frac{\sum_{j=1}^{n}(y_{(j)} - \hat{y}_j)^2}{n}}} = \frac{1}{r}\sqrt{\frac{\sum_{j=1}^{n}(y_{j(i)} - \hat{y}_j)^2 \times n}{\sum_{j=1}^{n}(y_{(j)} - \hat{y}_j)^2}} \tag{8}$$

Cook's distance is usually used for outlier detection where the outlier value deviates far from other independent variables' values. From Equations (3) and (8), $Dist(i)$ can be used to remove outlier or weak associative rules. For simplicity, we utilize Equation (3) in this paper.

## 3. Related Work

Mining customer preferences involves various parameters and decision support tools such as user preferences attributes, product specifications, and sentiments. Many related works focus on both user preferences and product specifications. The commonly used method includes cluster analysis, machine learning, genetic algorithm, and associative mining. The related works are shown in Table 1.

Clustering analysis groups preferences by performing clusters on the available data. Zhang et al. [18] consider product competitiveness when mining customer preferences. They proposed an information mining method that utilizes entropy and density-based clustering analysis to the customer preferences. Chong et al. [19] proposed a method to identify preference by clustering items based on multi-attributes such as consumer sensory scores. Seo et al. [20] proposed a recommender system for a group of items that are based on genre preference that may reduce clustering computation cost. Other clustering-based analyses were also proposed by Osama et al. [21] and Wang et al. [22].

**Table 1.** Summary of related works.

| Literature | Year | Clustering Analysis | Machine Learning | Genetic Algorithm | Collaborative Filtering | Associative Mining | Process Mining |
|---|---|---|---|---|---|---|---|
| Zhang et al. [18] | 2022 | √ | | | | | |
| Chong et al. [19] | 2020 | √ | | | | | |
| Seo et al. [20] | 2021 | √ | | | | | |
| Osama et al. [21] | 2019 | √ | | | | | |
| Wang et al. [22] | 2019 | √ | | | | | |
| Xiao et al. [23] | 2022 | | √ | | | | |
| Zheng et al. [24] | 2022 | | √ | | | | |
| Sun et al. [25] | 2021 | | √ | | | | |
| Aldayel et al. [26] | 2020 | | √ | | | | |
| Bi et al. [27] | 2020 | | √ | | | | |
| Gkikas et al. [28] | 2022 | | | √ | | | |
| Das et al. [29] | 2022 | | | √ | | | |
| Jiang et al. [30] | 2019 | | | √ | | | |
| Petiot et al. [31] | 2020 | | | √ | | | |
| Alhijawi et al. [32] | 2020 | | | √ | √ | | |
| Liu et al. [33] | 2022 | | | | √ | | |
| Liang et al. [34] | 2022 | | | | √ | | |
| Valera et al. [35] | 2021 | | | | √ | | |
| Fkih et al. [36] | 2021 | | | | √ | | |
| Davis et al.[37] | 2021 | | | | √ | | |
| Qi et al. [38] | 2022 | | | | | √ | |
| Tan et al. [39] | 2020 | | | | | √ | |
| Chen et al. [40] | 2021 | | | | | √ | |
| Ait-Mlouk et al. [41] | 2017 | | | | | √ | |
| Kaur et al. [42] | 2016 | | | | | √ | |
| Our Method | 2023 | | | | | √ | √ |

Machine learning can predict user preferences by learning from recorded transaction data. Xiao et al. [23] focus on the sentiment tendencies of the customer to extract their preferences. These tendencies are fine-grained to improve the performance of the analysis. The fine-grained sentiment analysis problem is converted into a sequence labeling problem to predict the polarity of user reviews. Since the problem involves sentiment analysis, the user-feature focus on the review dataset with text information, such as words with emotional words. Conditional Random Field (CRF) and neural networks were applied to analyze the text sequence. Zheng et al. [24] focus on immersive marketing and applied graph neural network models that consider essential attributes to improve the consumer shopping experience. Other related works related to machine learning were also proposed by Sun et al. [25], Aldayel et al. [26], and Bi et al. [27].

Genetic algorithms can find the most optimal preferences from various preferences patterns based on evolutionary algorithms. Gkikas et al. [28] proposed a combination of a method using binary decision trees and genetic algorithm wrappers to enhance marketing decisions. They focus on customer survey data to classify customer behaviors. As a model to classify customer behavior, Optimal decision trees are generated from binary decision trees, representing the chromosomes handled in the genetic algorithm wrapper. Das et al. [29] used a genetic algorithm to predict the premium of life insurance based on consumer behavior toward the insurance policies before and after-pandemic situations. Other work was proposed by Jiang et al. [30] and Petiot [31].

Collaborative filtering collects preferences from many customers and predicts a user's preferences. Alhiijawi et al. [32] applied a genetic algorithm with collaborative filtering to generate recommended preferences using multi-filtering criteria. Liu et al. used weighted attribute similarity and rating similarity in collaborative filtering that can alleviate data sparseness. Liang et al. [34] focus on diversifying recommendations using neural collab-

orative filtering that can achieve high diversity in a recommendation. Valera et al. [35] proposed a method that uses collaborative filtering not for single-user preferences but for group preferences. The method takes into account taking individual preferences and context in the group. Other work includes Fkih et al. [36] and Davis et al. [37].

Associative mining extracts associative rules from available data to recognize patterns based on basket analysis. Qi et al. [38] proposed an algorithm that utilized weighted associative rules to discover frequent item sets with high values. Tan et al. [39] proposed top-k rules mining based on MaxClique for contextual preferences mining. In the method, they applied the problem to association rules extracted from preference databases. They offered a conditional preference rule with context constraints to model the user's positive or negative interests. Other work includes Ait-Mlouk et al. [41] and Kaur et al. [42].

The given related works focus only on the interestingness of an item, such as buyer sentiments towards one attribute, i.e., rating or prices. However, it is hard to simultaneously extract implicit preferences with multi-variate attributes such as price, rating, and discount. There exists a trade-off between choices and attributes of target items. Moreover, it is hard for users to decide on many multi-variate attributes simultaneously. Therefore, there is a need to balance between choices and attributes in customer preferences.

## 4. The Problem of Implicit Preference Extraction

First, we formalized the problem of extracting service preference from the service model representing the business rule based on the sales log. First, we define preference as follows:

**Definition 1** (Preference). *A preference $\sigma$ is denoted by n-tuple $(\alpha_1, \alpha_2, \cdots, \alpha_n)$ where $\alpha_n$ is called as preference attribute of $\sigma$.*

We formalize a problem which is to achieve a goal for service preference extraction.

**Definition 2** (Service Preference Extraction Problem).
**Input:** *Sales log $S$ containing items $i_1, i_2, \cdots, i_n$, explicit preferences set $P = (\alpha_1, \alpha_2, \cdots, \alpha_n)$*
**Output:** *Implicit preferences set $P' = (\beta_1, \beta_2, \cdots, \beta_n)$*

Based on the problem definition above, we input a sales log $S$ and explicit preferences $P = (\alpha_1, \alpha_2, \cdots, \alpha_n)$. The sales log contains items $i_1, i_2, \cdots, i_n$. The preferences $\alpha_1, \alpha_2, \cdots, \alpha_n$ corresponds to each item $i_1, i_2, \cdots, i_n$. Example of explicit preferences $P$ and implicit preferences $P'$ can be given as $P = (High, Low, No-Interest)$ and $P' = (High, Low, High)$ where abstract attributes value such as *High*, *Low* and *No-Interest* corresponds to the preference of item $(Price, Rating, Discount)$. Here, the implicit preference reveals *No-Interest* in attribute *Discount* can be replaced with *High* value.

First, we extract implicit preferences set $P' = (\beta_1, \beta_2, \cdots, \beta_n)$ then output the new preference $P'$ to represent the choices of items $i_1, i_2, \cdots, i_k$ that have strong correlation between attributes. The implicit preferences do not satisfy $\beta_n = \alpha_n$ for all of its elements. If $\beta_n = \alpha_n$, then the explicit preference for item $i_n$ does not change. However, if there is at least one element that satisfies $\beta_n \neq \alpha_n$, we can call $P' = (\beta_1, \beta_2, \cdots, \beta_n)$ as implicit preference. Therefore, we define implicit preference as follows:

**Definition 3** (Implicit Preference). *For a given item set $I = \{i_1, i_2, \cdots, i_n\}$ and its explicit preference $\sigma = (\alpha_1, \alpha_2, \cdots, \alpha_n)$, implicit preference is defined as $\pi = (\beta_1, \beta_2, \cdots, \beta_n)$ where $\alpha_n$ and $\beta_n$ satisfy the following:*
*(i)   There exists at least one $\beta_n \neq \alpha_n$ such that each $\alpha_n$ and $\beta_n$ represents the preference of item $i_n$.*
*(ii)  Item set $I_{\mathcal{E}} \subseteq I$ for explicit preference $\sigma$ and item set $I_{\mathcal{I}} \subset I$ for implicit preference $\pi$ satisfy $(I_{\mathcal{I}} \subset I_{\mathcal{E}})$.*

Definition 3 denotes implicit preference $\pi = (\beta_1, \beta_2, \cdots, \beta_n)$ that satisfies $\beta_n \neq \alpha_n$, respectively. Implicit preferences are regarded as preferences that are not shown in explicit preferences. However, no implicit preferences are extracted if $\beta_n = \alpha_n$ holds for all elements.

We use an online ordering system as an example to demonstrate our ideas and approach. The online ordering system takes orders for a Thai cuisine menu. The menu can be modeled with Petri net as shown in Figure 2. Figure 2 is represented by a Petri net that shows a selection of items from the menu in a sequence form from left to right. Each node and arc shown in the model represents a possible combination of the items. The menu contains eight categories, which are *Appetizer1*, *Appetizer2*, *Soup*, *MainCourse1*, *MainCourse2*, *Dessert1*, and *Dessert2* is shown in Figure 2 as a group of connected nodes and arc from left to right. The combination of items can be decided by looking at each attribute, i.e., *Price*, *Rating*, *Calorie*, *Discount*. The customer can choose to select one from each category as their choice from the course menu. The menu shows the restaurant's price, rating, calories, and discounts. The combination of selections that the user can make is 192,400 combinations. Therefore, it is hard for users to decide on the course that suits their preferences.

For further explanation, we give an example of three customers with different preferences. Customers usually express their preferences intuitively. We can conclude that customers make decisions based on specific attributes, but it is hard to look into the details of attribute values, especially for first-time customers. Moreover, they usually depend on the server or service advisor for recommendations. Therefore, the server or service advisor must make suitable recommendations that satisfy the customer. Let us consider the preference of the following customers:

1. *Alice*: She prefers food with low calories and prices. She will usually ask, "I prefer a healthy diet. What is the food with low calories but not too expensive." However, she might be implicitly attracted to try highly-rated food or discounted menu.
2. *Bob*: He prefers food with a less expensive and good rating. Therefore, he will ask, "What is the best affordable food you recommend?". It seems he does not care about calorie consumption and is also not interested in discounted food.
3. *John*: He has no specific preference. Therefore, he will ask, "What is today's special?" or "What is your recommendation?".

Since *Alice* prefers low-price and low-calorie, we can offer a new menu that allows the customer to choose items that strongly relate to low-price and low-calorie, for example, *Grilled Shrimp* for appetizers are low cost and always requested with *Pudding* because it has low calories. However, because only appetizers and desserts have a frequent pattern for a low price and low calories, other types of items more explanation, we'd like to give you Jued for soup and *Kanoom Jeen Namya*, will also be requested by the customer because they have attributes with low price and low calorie. In the case of Bob, he prefers low-price but good ratings. Similarly, he might be attracted to calorie and discounted food if he looks into the details. In the case of John, it is the hardest to meet his demands since he also needs to know what he prefers so that he will set all his preferences as *No-Interest*.

Customers with explicit preferences will be restricted to a few uninteresting choices. Fewer choices may reduce repeat customers. The case is similar to Bob, who prefers inexpensive food with high ratings where he does not care about calorie consumption and discount. Sometimes, customers such as John do not know what he likes. Therefore, to respond to preferences that are not specific, the customers should be given a range of attractive selections on the menu that might satisfy their implicit preferences.

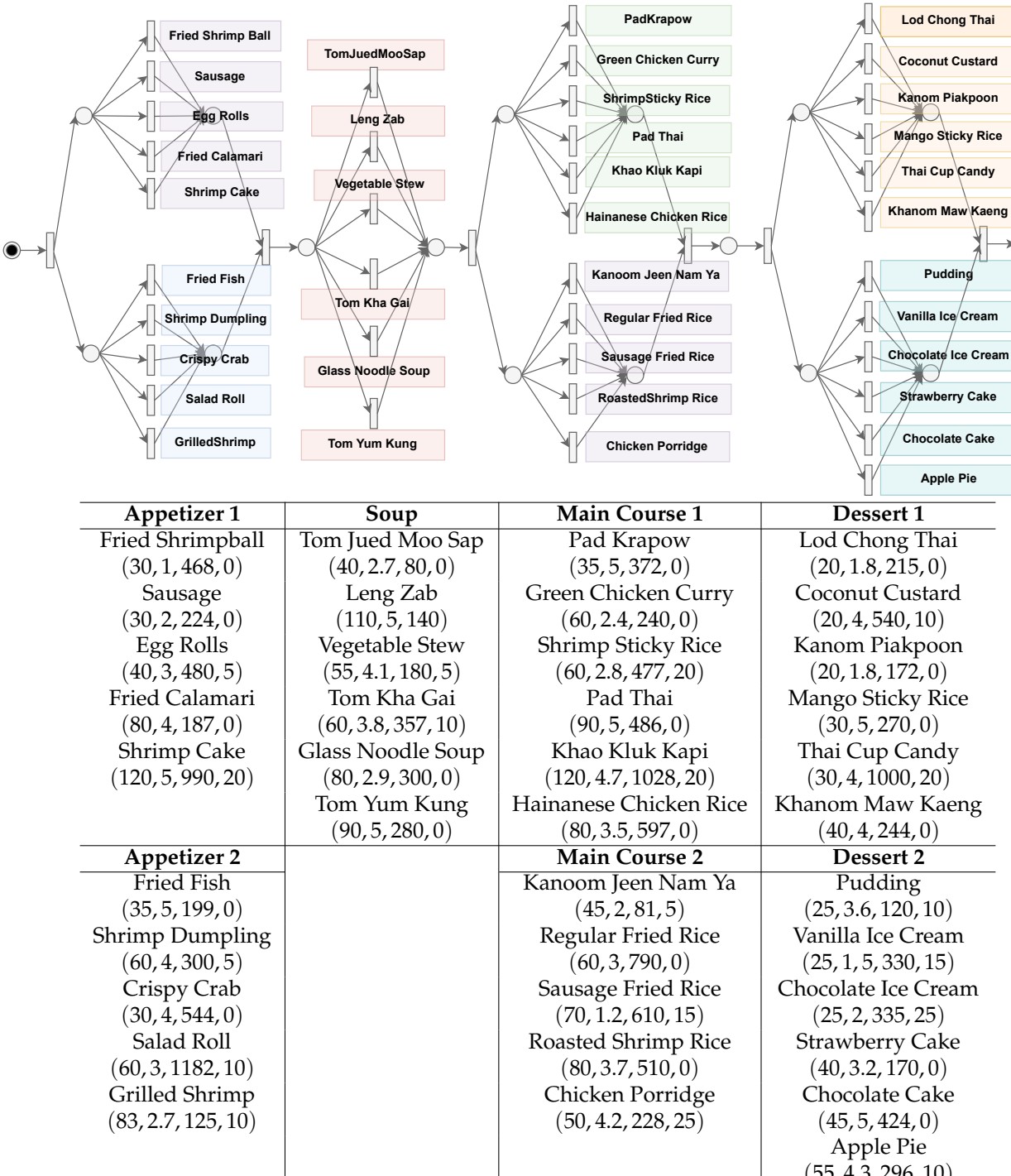

**Figure 2.** Menu of items with their attributes (*Price, Rating, Calorie, Discount*).

## 5. Preference Refinement by Associative Rule and Process Mining

We illustrate the details of our method in Figure 3. First, we extract frequent items that include associative rules with low confidence and lift value using Apriori. Then, we remove irrelevant rules that are (i) duplicate rules, (ii) not satisfying user preferences, and (iii) outlier rules in which correlation distances are too high.

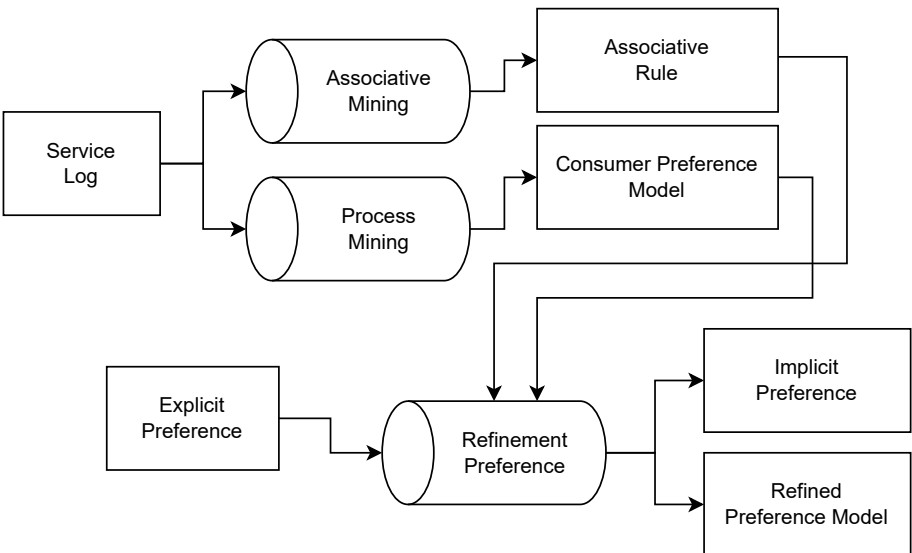

**Figure 3.** Our approach. We used data mining and process mining to extract the customer behavior model.

Figure 4 shows the overview of our approach. Given a preference *P* of some attributes such as *Attribute 1*, *Attribute 2*, and *Attribute 3*. For example, *Price*, *Calorie* and *Rating*. These attributes as given as (*Low*, *High*, *No−Interest*) where each represents the preference of *Price*, *Calorie*, and *Rating*. The value can describe a range of values, such as 0 to 100. For example, if the value is 0 to 50, then the value is represented by *Low*; if the value is between 51 to 100, then the value is represented by *High*. If no preferences are given, then the value can be between 0 to 100, which *No−Interest* represents.

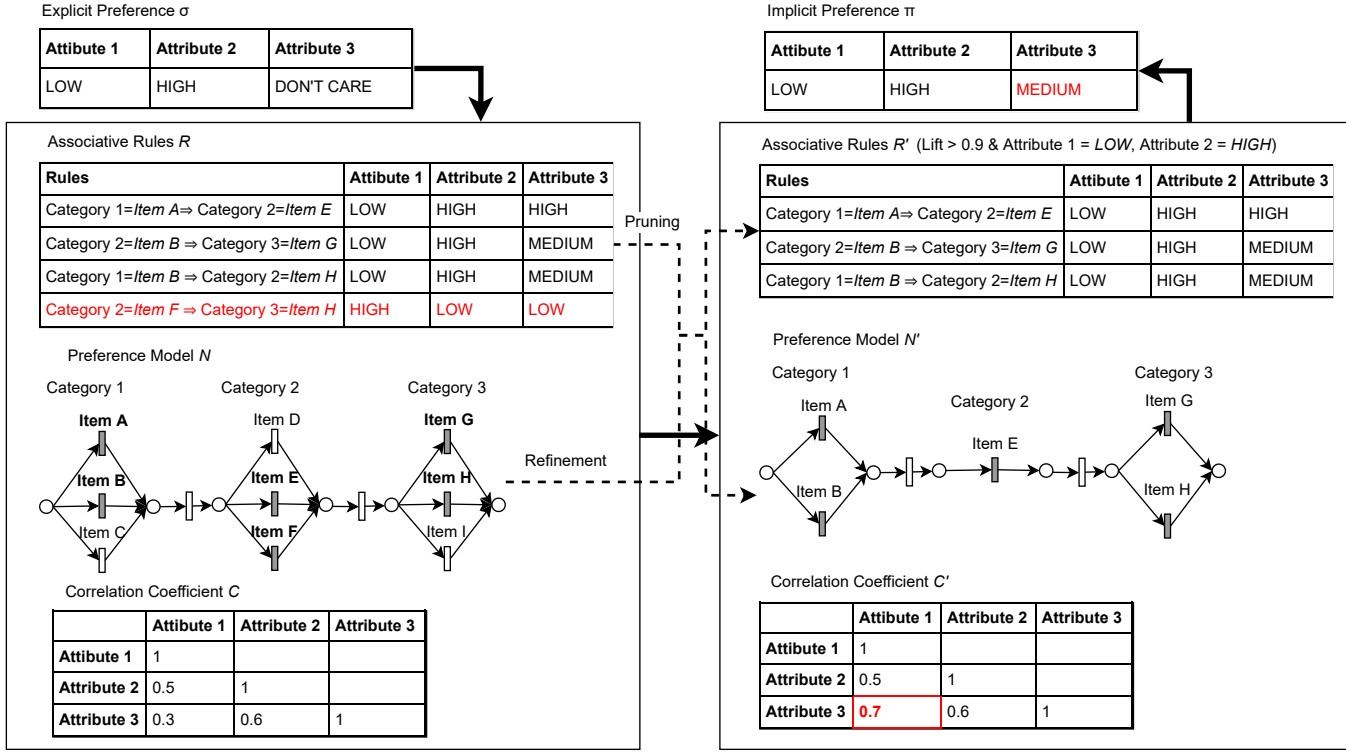

**Figure 4.** The detailed we added the explanation of red part overview of our approach. The input is explicit preference $\sigma$, and the output is implicit preference $\pi$. Both preferences are bounded by attributes and items in associative rules *R* and preference model *N*. The red part shows preferences that adapt to changes.

From here, the preferences are taken as a reference when extracting the associative rules from the service log. The associative rules can be represented such as $(Category\ 1 = Item\ A) \Rightarrow (Category\ 2 = Item\ E)$. *Category* 1 is the category of item that corresponds to the general options available in the services. For example, *Appetizer*, *Main Dish*, *Soup*, and, *Desert*. Each item, such as *Item E*, represents the selection within the category. For example, for *Appetizer*, some items such as *Item E* are available for selection. The rules are accompanied by a set of attributes with values calculated from the average of the attributes of each item found in the rules. For example, for rules $(Category\ 2 = Item\ F) \Rightarrow (Category\ 3 = Item\ H)$, the preference of attributes found in the rules is $(High, Low, Low)$.

Next, the correlation between attributes is calculated. Based on the correlation values, i.e., Pearson correlation [43], we filter out some rules with attributes that have a high distance value using Cook's distance [13]. Then, we recalculate the correlation and value of attributes of the rules. Since the rules with high correlation were pruned, we can obtain the new preferences $P'$ that are optimized for the customer. The changes are then used for the refinement of the service workflow. The new service workflow represents a workflow that satisfies the new preference $P'$. The rules which satisfy Lift $\geq 0.9$ are preserved. Associative rules with a Lift value that is larger than 1 show a strong relationship between an item set $X$ and $Y$. Rules with Lift $\geq 1$ are considered strong. Note that the value is not absolute; some rules with a high lift value do not always have higher confidence than those with a lower lift value. If we increase the threshold of the Lift value, the number of extracted rules will be reduced. Therefore, we recommend reducing the Lift value to Lift $\geq 0.9$.

The next step is to find out which menu the customer prefers. We set the parameters within a specific range to identify the relationship between items. For example, for *Price* attribute, we set the range between 0.0 to 80.0 as *Low*, and 81.0 to 140.0 as *High*. This range can be decided using discretization such as the binning method [44] or pre-defined by the user. We call the set of values as preference class $\mathcal{X}$.

**Definition 4** (Preference Class). *For a given attribute value $\alpha$, the value of $\alpha$ can be represented with preference class $\gamma$ if $\alpha$ ranges between $[u, v]$ denoted by $(\gamma, [u, v])$.*

For example, we separate *Low* and *High* values based on the median value. Therefore, we can give the parameters of preference $\alpha$ based on preference class $\mathcal{X}$ as follows:

(i)　　*Price* : $(Low, [0, 80]), (High, [81, 140])$
(ii)　　*Rating* : $(Low, [0, 3]), (High, [3.1, 5])$
(iii)　　*Calorie* : $(Low, [0, 600]), (High, [601, 1040])$
(iv)　　*Discount* : $(Low, [0, 10]), (High, [15, 30])$
(v)　　*No-Interest* : $(NI, [0, \infty))$

As a practical method to capture customers' intuitive preferences, Based on the range of values, we allow setting the preferences intuitively rather than giving specific values.

First, we extract the set of associative rules $R$ from the sales logs. A threshold of more than 0.9 is used for Apriori. Next, we extract the customer process model using process mining. The process model represents the ordering sequence of items shown in the sales log. Then, all rules with a low Pearson's correlation coefficient, i.e., less than 0.5 are removed. For rules $r_i$, we check the regression. If the Cook's distance of $r_i$ is more than the threshold $h = 1.5$, we remove the rule $r_i$. The reason for setting the threshold value of 1.5 times is that most distance that is too high exceeds 1.5 times more said to deviate from the average value. However, depending on the analysis of the services, the user must decide on a suitable value.

The procedure for extracting implicit preference is given in Procedure ≪Procedure of Implicit Preference Extraction≫. The procedure discovers the process model $N$, and extracts the set of associative rules $R$. Then the procedure filters out weakly correlated associative rules and items in preference model $N$ and the set of rules $R$ using Cook's distance. Table A2 shows the rules that satisfy the explicit preferences, and the Pearson correlation value is shown for each relation between attributes *Price-Rating* (PR),

*Price-Calorie* (PC), *Price-Discount* (PD), *Rating-Calorie* (RC), *Rating-Discount* (RD), and *Calorie-Discount* (CD). The table also shows the number of rules extracted. We only focus on preferences that can extract more than 30 variations of rules. Note that the set of rules also includes duplicated rules at this phase. From the set of rules, we filter out the weak rules that have weak correlation values.

We utilize Cook's distance and associative mining. Cook's distance characteristic is that during regression, it can detect highly influential items from a set of items. The measurement is regarded as the value of the influence of an item if it is removed from the group. In Cook distance, we can control how large the influence can be tolerated for an item by maintaining the sensitivity of the distance. For example, as a rule of thumb, a distance 1.5 times above the mean indicates that an item correlates less with other items in the group. Depending on the situation, the user can adjust the control parameter. Cook's distance is a distance based on regression and is suitable for multivariate analysis. Therefore, we handle all items as a highly correlated group with high sensitivity between them. Items that are far from the group have more distance. In Cook's distance, we perform a least-squares regression analysis to measure the influence of one item on another item in the same group.

Figure 5 illustrates the overview of removing weak correlated multivariate associative rules with our method. The rule is removed by averaging the value of $\alpha_1, \alpha_2, \cdots, \alpha_i$ when $\alpha_i$ is removed from the attributes observation $i$. Value $u$ and $v$ represent the value of each rule shown as $\circ$. The dotted line between $\circ$ and the regression line (solid line) shows the residual of the regression. The absolute difference $(\hat{y}_{j(i)} - \hat{y}_j)^2$ is averaged by regression MSE using the residual value. Implicit preference $\Pi$ can be produced from the difference of explicit preference before and after the removal such that $\Pi = \sigma - \sigma'$.

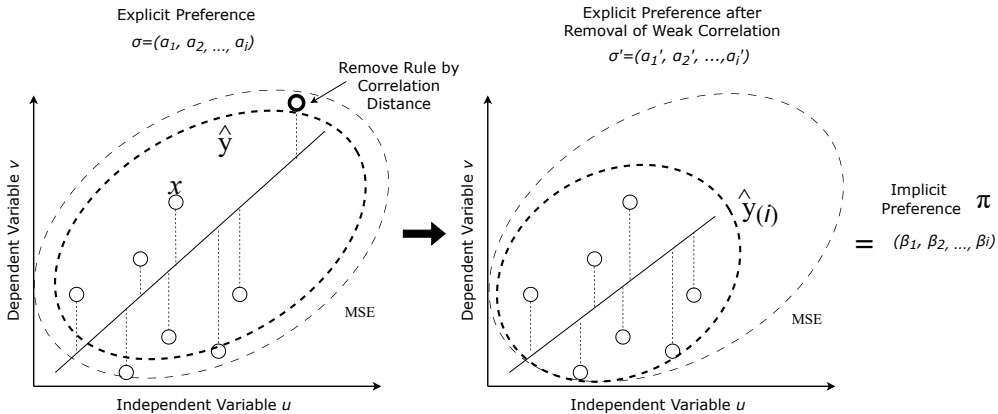

**Figure 5.** Overview of removing weak correlated multivariate rules and how residual difference $\hat{y}_{(i)} - \hat{y}$ can produce implicit preference $\Pi$.

Next, to calculate the cut-off point, we utilize Cook's distance, denoted by $Dist(r_m)$ as shown in Equation (3). We calculate the correlation distance if the set or extracted associative rules is not an empty set such as $R \neq 0$. Therefore, the equation can be given as in Equation (9). $cutoff(Dist(R))$ is the adjusted mean value for the cut-off point of the correlation distance.

$$cutoff(Dist(R,h)) = \begin{cases} h \times \frac{1}{m} \sum_{i=1}^{n} Dist(r_m), & \text{if } R \neq \phi, r_m \in R \\ 0, & \text{otherwise} \end{cases} \tag{9}$$

The value of $mean(Dist(R))$ denotes the mean of the correlation distance calculated by Cook's distance. The sensitivity of the outlier removal is controlled by variable $h$, which is the distance ratio from the mean value that is calculated from $Dist(r_m)$ using Cook's distance formula shown in Equation (3).

Algorithm 1 shows our main procedure in extracting the implicit preference. The complexity of the Algorithm 1 is $O(|R||\mathcal{A}| + |R|^2 + |T||R|)$ where $|R|$ is the number of

rules, $|\mathcal{A}|$ is the number of preference attributes and $|T|$ is the number of action labels. The ≪Procedure of Implicit Preference Extraction≫ shows the whole procedure using process mining, associative mining, and implicit preference extraction using Cook's distance.

≪Procedure of Implicit Preference Extraction≫

Input: Sales Log $S$, set explicit preference $\Sigma = (\sigma_1, \sigma_2, \cdots, \sigma_m)$, lift threshold $\mathcal{T}$

Output: Refined Preference model $N'$ and set of implicit preference $\Pi = (\pi_1, \pi_2, \cdots, \pi_n)$

1°    Discover process model $N = (P, T, A)$ from order log $E$ using process mining.
2°    Extract associative rules $R = \{r_1, r_2, \cdots, r_n\}$ where $r_n = (X, Y, \mathcal{A})$ from event log $E$ which satisfies Lift $\geq \mathcal{T}$.
3°    Extract implicit preference $\Pi$ for $N$ using Algorithm 1.
4°    Output refined process model with implicit preferences $(N', \pi)$ and stop.

---

**Algorithm 1:** IMPLICIT PREFERENCE EXTRACTION

---

**Input:** Preference Model $N = (P, T, A)$, Explicit preference $\sigma = (\alpha_1, \alpha_2, \cdots, \alpha_m)$, Cut-off threshold $h$, Set of associative rules $R$

**Output:** Refined Preference model $N'$ and implicit preference $\pi = (\beta_1, \beta_2, \cdots, \beta_n)$

1:   $R' \leftarrow \varnothing, \delta \leftarrow 0$
2:   **for** each $r_i \in R$ **do**
3:     **for** each $\alpha_m \in \mathcal{A}$ of $r_i$ **do**
4:       **if** $\alpha_m \simeq \sigma_m$ **then**
5:         $R' \leftarrow R' \cup \{r_i\}$ ▷ *Add $r_i$ to new set of rules $R'$ if $\alpha_m$ satisfies $\sigma_m$*
6:       **end if**
7:     **end for**
8:   **end for**
9:   **for** each $r_i \in R'$ **do**
10:     **for** each $r_j \in R' (i \neq j)$ **do**
11:       **if** $r_i : X \Rightarrow Y$ and $r_j : Y \Rightarrow X$ **then**
12:         $R' \leftarrow R' - \{r_i\}$   ▷ *Remove duplicated rules*
13:       **end if**
14:     **end for**
15:     **if** Cook's distance $Dist(r_i) \geq cutoff(Dist(R', h))$ **then**
16:       $R' \leftarrow R' - \{r_i\}$   ▷ *Remove outliers with low correlation*
17:     **end if**
18:   **end for**
19:   **for** each task label $t \in T$ **do**
20:     **for** each $(r_i : X \Rightarrow Y) \in R'$ **do**
21:       **if** $t \notin X$ or $t \notin Y$ **then**
22:         $T' \leftarrow T - \{t\}$ ▷ *Remove label $t$ from $N$ if $t$ is not in item sets $X$ or $Y$*
23:         $N' \leftarrow N(P, T', A)$ ▷ *Create new process model $N'$ with $T'$*
24:       **end if**
25:     **end for**
26:     **for** each $\alpha \in \sigma$ **do**
27:       $\delta \leftarrow mean(T', \alpha_m)$ ▷ *Calculate mean of attribute $\alpha_m$ for each $t \in T'$*
28:       $\beta_m \leftarrow prefClass(\delta, \mathcal{X})$ ▷ *Set $\beta_m$ with value $\delta$ by mapping to preference class $\mathcal{X}$*
29:       $\delta \leftarrow 0$
30:     **end for**
31:     $\pi \leftarrow (\beta_1, \beta_2, \cdots, \beta_m)$ ▷ *Construct the implicit preference $\pi$*
32:   **end for**
33:   **return** $(N', \pi)$   ▷ *Output refined process model with implicit preferences and stop*

---

The proposed procedure utilizes the mechanism of Cook's distance when removing one variable $\alpha$ from the observation. For example, for a given preference on *Price, Rating, Calorie*, and *Discount*, we can observe the influence of *Price* by setting *Price* as target $i$. We can observe the improvement of residual coefficient $R^2$ in the regression. Figure 6a

shows the distance for each item of $n = 1000$ rules. The cutoff line shows the threshold at $4/n$. Figure 6b compares residuals before and after removing weakly correlated rules. The $R^2$ value was 0.691, but the normal distribution distorted slightly to the positive value. Figure 6b shows the residual coefficient improved to 0.736. Figure 6b shows that the correlation improved because the distribution changed from a more dispersed distribution into a tighter distribution resulting in a higher $R^2$ value. The green dots represent the distribution after weakly correlated rules were removed, and the blue dot shows the distribution before the removal. We can see the blue dots are the outliers that were removed from the set of rules. Here, we confirmed that Cook's distance is effective in our procedure.

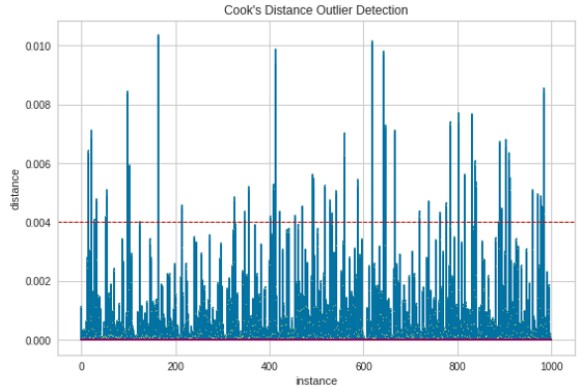

(**a**) Cook's distance-based cutoff for attribute $\alpha_i$.

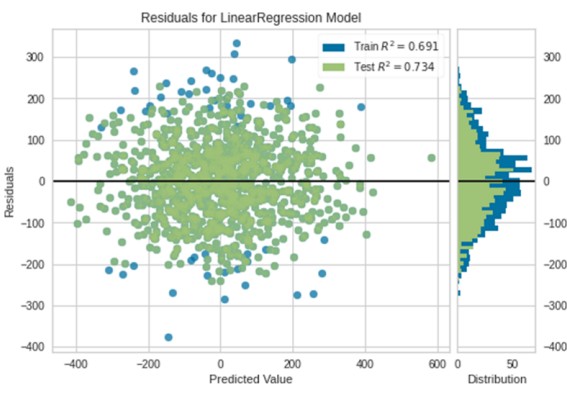

(**b**) Residual plot comparison.

**Figure 6.** The effect on residual after using Cook's distance. The residual coefficient improved after removing weakly correlated rules.

## 6. Application Example and Evaluation

We applied our method to an order transaction that included 40 items, as shown in Figure 2. As stated in Section 4, customers can have at most 192,400 combinations of unique orders. Here, we evaluated the data with at least 60,000 data recorded in the sales log. Based on the steps in Procedure 1, we perform data cleaning to remove duplicate rules. We filter rules using Cook's distance to preserve rules with strong correlations. The procedure will remove rules that exceed the threshold value such that $\mathcal{T} > 1.5$. Figure A1a,b shows the detection of irrelevant rules which are over the threshold value. Each figure shows *Price*, *Rating*, and *Calorie* as independent variables. The same procedure also applies to *Price* and *Discount*. Cook's distance is calculated by taking the independent variables from the observation. The figures show that the Cook's distance of index $i$ that is over the red line corresponds to rule $r_i$ will be removed. For example, Figure A1a show that rules $r_6, r_7, r_{14}$, and $r_{15}$ were removed from the set of rules R. Figure A1b,c also shows the removal of rules that exceeds the threshold value.

Apriori extracted 40 associative rules. The procedure outputs 11 rules with strong correlation such as *Price-Rating* and *Calorie-Discount*. From the result, the preferences for low prices and low calories strongly correlate with ratings and discounts. By successfully identifying this factor, we can motivate the customer to decide on this menu by offering more selections with a high rating and discount. The improvement of correlation, i.e., preference $p_{10}$ is shown in Figure 7a,b. After applying our method, the figures show the improved correlation of preference $p'_{10}$.

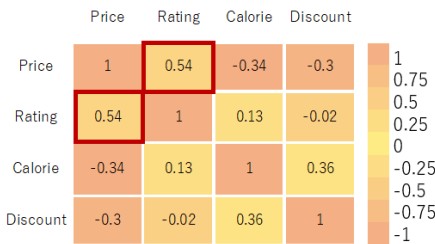

(**a**) Correlation matrix of preference $P_{10}$.

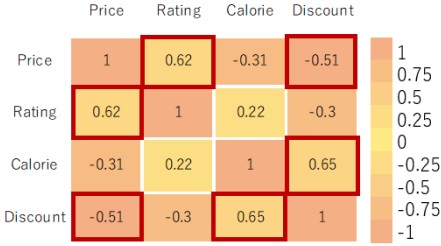

(**b**) Correlation matrix of preference $P'_{10}$.

**Figure 7.** Heat map of correlation matrix before and after applying Procedure 1.

The pruned and optimized preferences result is shown in Table 2. For a given preference with high correlation, 6 preferences ($p_0$, $p_2$, $p_6$, $p_{10}$, $p_{14}$ and $p_{19}$) were extracted. The refined preferences is shown as $p'_0$, $p'_2$, $p'_6$, $p'_{10}$, $p'_{14}$ and $p'_{19}$. The course menu selection is optimized as shown in Figure 8a,b.

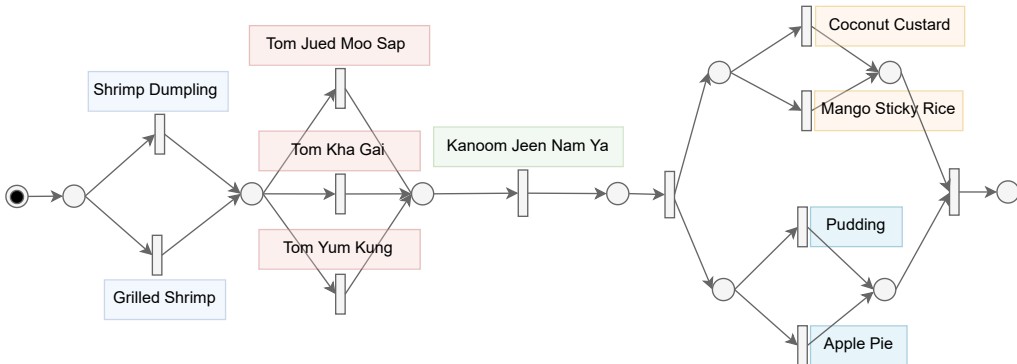

(**a**) Preference model $N_{(Low,NI,Low,NI)}$ refined for low price and low calorie.

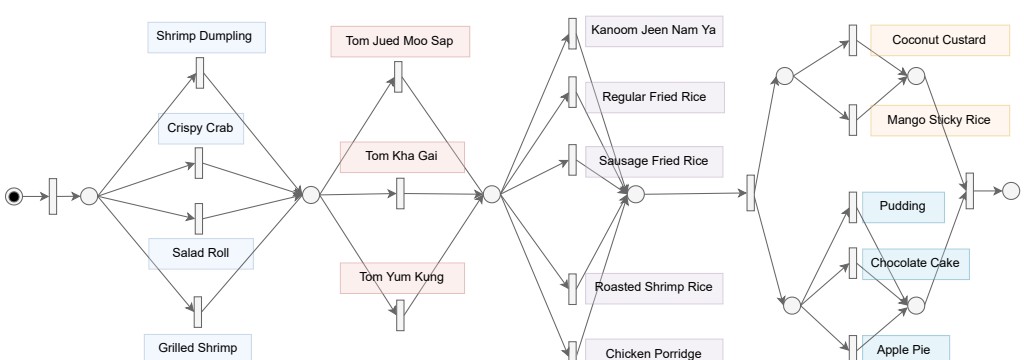

(**b**) Preference model $N_{(Low,High,NI,NI)}$ refined for low prices and high rating.

**Figure 8.** Implicit preference models output by Procedure 1.

Next, we evaluate our approach. First, we calculate the correlation coefficient rule removal based on Cook's distance. From Table A2, the value for the correlation coefficient of rules $p_0$, $p_2$, $p_6$, $p_{10}$, $p_{14}$ and $p_{19}$ was around 0.54. Figure A1 shows the removal of rules with a distance that exceeds the threshold value for $p_{10}$. One independent variable is removed from each observation. Around 20% of the total rules were removed. Here, we found that preferences with *No-Interest* (NI) reveal implicit preference with the highest correlation. For example, in $p_{10} = (Low, NI, Low, NI)$, *Rating* and *Discount* was set to *No-Interest*, but $p'_{10} = (Low, NI, Low, High)$ shows the preference have strong relation to high discount.

**Table 2.** Explicit preference (before) and implicit preferences (after).

| Old Preference (Explicit) | | Refined Preference (Implicit) | |
|---|---|---|---|
| $p_0$ | (NI, NI, NI, NI) | $p'_0$ | (Low, NI, NI, Low) |
| $p_2$ | (NI, NI, Low, NI) | $p'_2$ | (NI, NI, Low, Low) |
| $p_6$ | (NI, High, Low, NI) | $p'_6$ | (NI, High, Low, Low) |
| $p_{10}$ | (Low, NI, Low, NI) | $p'_{10}$ | (Low, NI, Low, High) |
| $p_{14}$ | (Low, High, NI, NI) | $p'_{14}$ | (Low, High, NI, Low) |
| $p_{19}$ | (Low, High, Low, NI) | $p'_{19}$ | (Low, High, Low, Low) |

The mining result shows that *Calorie-Discount* and *Price-Discount* show an increase in correlation from 0.4 to 0.7 and −0.3 to −0.5. Here, the price and calorie attributes have high correlations to discounts. Based on this information, the seller of course menu can focus on discounted items for low prices and low calories. Preference $p_{10}$ can be refined as $p'_{10}$ as $(Low, NI, Low, High)$. In the case of $p'_{10}$, low-price items and low-calorie item menus usually attract customers that prefer highly discounted menus.

Figure 9 shows the correlation coefficient $\mathcal{R}$ for *Price-Calorie* (PC), *Rating-Calorie* (RC), and *Rating-Discount* (RD). *PR'*, *PC'*, *PD'*, *RC'* and *RD'* is the correlation after applying our method. Most of the attributes' correlation increased. Here, some of the correlation changes from a negative correlation to a positive correlation. The correlation between negativity and positivity does not give much meaning other than the increase and decrease relationship of either attribute. In this paper, we focus on the strength of the correlation regardless of the negativity and positivity of the correlation.

Figure 10a shows the comparison of the correlation coefficient between attributes. The *Calorie-Discount* relation shows the most improvement in preferences $P_0$, $P_2$, $P_6$, and $P_{10}$. *Rating-Discount* and *Price-Discount* show significant improvement in preference $P_{10}$ and $P_0$. This is because $P_0$ do not specify any interest where all attributes were set to no-interest (*N/I*) and $P_{10}$ is almost similar to $P_0$ because the attribute values were set to *Low* and *No-Interest*. *Price-Calorie* shows significant improvement in preferences $P_0$, $P_2$, and $P_6$. In contrast, *Price-Rating* and *Rating-Calorie* correlation shows an improvement. From here, we can conclude that *Price-Rating* and *Rating-Calorie* are explicitly expressed in the preferences. Therefore, relations such as *Calorie-Discount* should be considered when recommending a new menu. Figure 10b shows the comparison for the average improvement of the coefficient for each preference. From the result, preferences with more no-interest specifications show the most improvement. This shows that our method can extract implicit preferences. However, depending on the pattern of preferences of customers, the preferences with more specifications, such as *High* and *Low*, may have different improvement values. This may be caused by the variations of selections and combinations allowed in the services, i.e., the business rules.

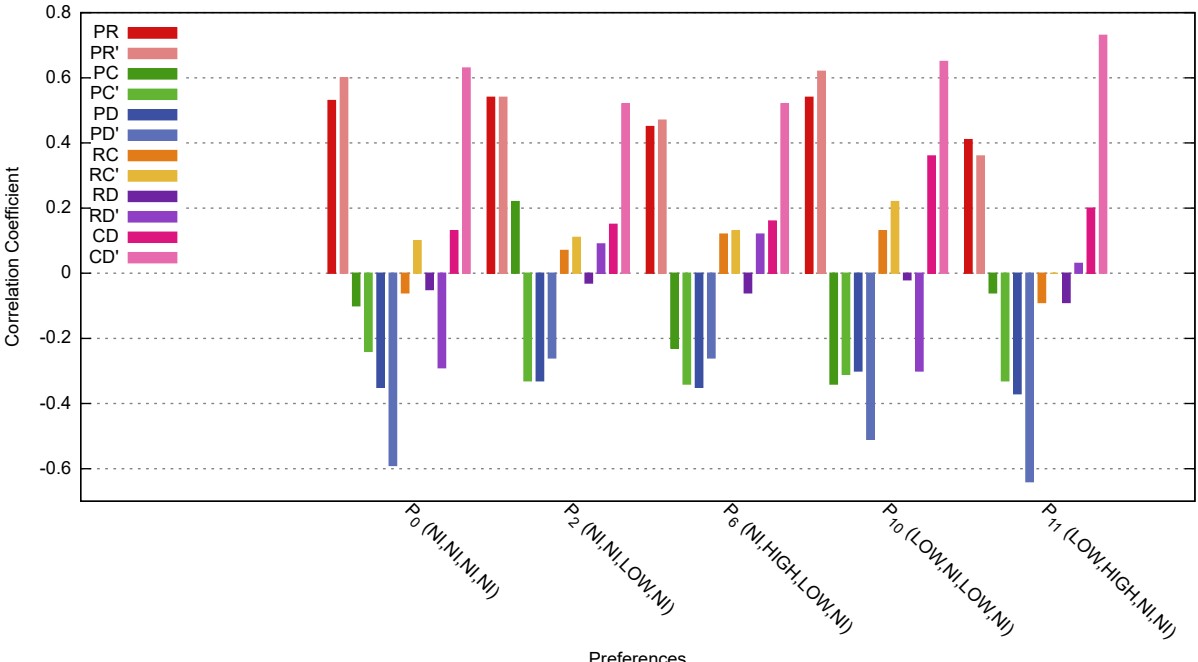

**Figure 9.** Comparison for the attribute's correlation coefficient for each preference.

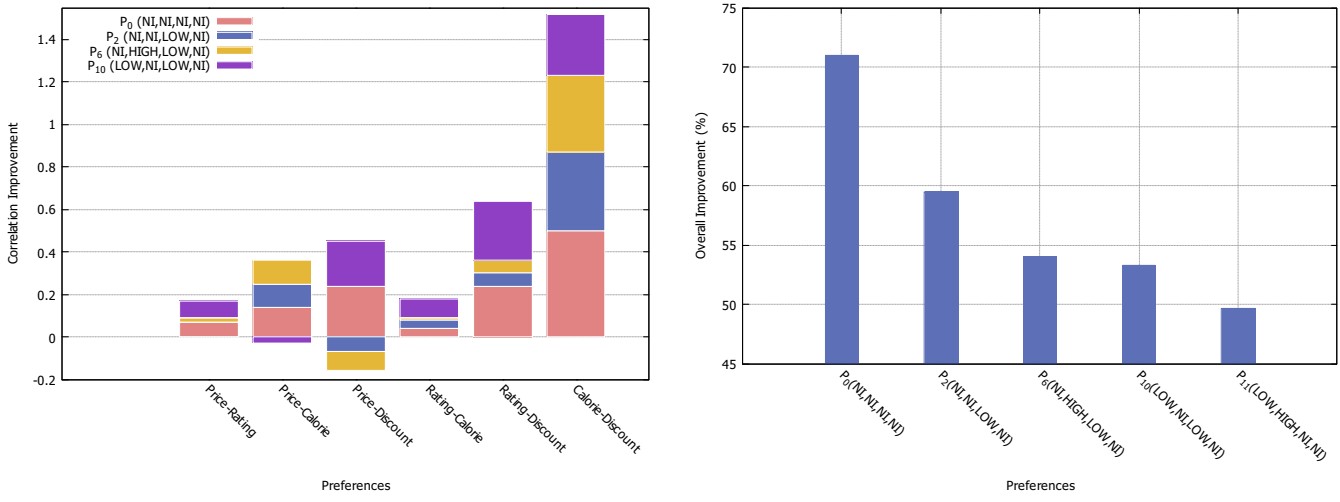

(**a**) Improvement of correlation coefficient.      (**b**) Average improvement.

**Figure 10.** Improvement result. Calculated based on the correlation of implicit preferences.

## 7. Discussion

Previous research focuses on finding explicit preferences from existing data such as product specifications, sales logs, reviews, and surveys. Explicit preference can be easily extracted from existing data mining methods such as clustering analysis, machine learning, associative mining, and genetic algorithm. The explicit preference can be gathered from the mining result. However, the gap lies when extracting implicit preference. Implicit preference in our study is regarded as an 'unknown preference' by the user. It means that the customer does not know their true preference unless some relation related to their explicit preferences are given. In this method, we use a combination of process mining and associative mining to extract implicit preferences. The ultimate benefit is that even if they do not give any explicit preference, which is 'no preference' set for all attributes of an item, the method can still extract what they might prefer by referring to the previous preference

of the previous customer as a starting point in making a decision on their preferences. Intuitively, the user requires less effort but more flexibility in making decisions.

The proposed method's main characteristic is to partially extract implicit preferences where specific preferences for certain attributes related to a service or product are not the same. For example, in the given menu, course attributes such as *Price*, *Rating*, *Discount*, and *Calories* are four attributes the customer must consider. Therefore, even if they have a certain preference, such as low price and high rating, the approach can find implicit preferences that are most likely to be preferred by the user, such as high discount and low calorie. By converting numerical values in the attributes into abstract values such as category, i.e., *High* and *Low*, we can intuitively present the result to the customer. From the example, most restaurant managers generally refer to the best-selling items to improve their menu. However, the best-selling item is independent of each other, and sometimes there is no reason why such best-selling items perform better than other items. We assume that best-selling items usually is driven by other items but with less frequency. We can extract related items (mostly in sequence) from process mining to be selected by frequency. In addition, associative mining strengthens the correlation with support and lift value.

The main difference with previous works that focus on extracting explicit and implicit differences is in combination with the process mining model. Nguyen et al. [45] focused on integrating explicit and implicit ratings by using latent factors in the recommendation system. They proposed two models, which is for experienced user and inexperienced user. The method is solely based on the user's rating of each product or service item. Vu et al. [46] proposed a method to analyze and reveal implicit references using the probabilistic method. The method depends on exploratory analysis and a large group of samples to accurately extract implicit preferences.

In our process mining model, i.e., Petri net, we can explicitly represent relations between items from process mining and compare the sequence of items with associative rules. In our approach, our method emphasizes relations based on combinations of sequentially connected choices. It is optimized based on the frequency of items identified by process mining and the frequency identified by associative rule.

In extracting highly correlated associative rules, we utilized Cook's distance. The Cook's distance can be given in two versions; the $L_1$-norm (Manhattan distance-based) shown in Equation (3) and $L_2$-norm (Euclidean distance-based) version shown in Equation (5). According to Aggarwal et al. [47], the difference in $L_p$-norms is that due to the high dimensional data, $L_1$-norm is constantly preferable. The larger the value of $p$, the less qualitative meaning a distance metric holds. In our research, we perform multivariate data mining where the scale for each attribute in data variables differs. Therefore, Cook's distance in Manhattan-based form is preferable compared to Euclidean-based form due to sensitivity and scale in terms of the value of observation $y$.

We highlight the advantages of our method in extracting implicit preferences. At first, a customer intuitively set their preferences. In our method, we regard the preferences as explicit preferences. Implicit preference is a preference that the customer does not know. In general, even a service user only knows their preference once they experience using the service. This is most likely to happen to first-time users. Therefore, by extracting previous users' experiences, we can extract the options of services most likely to be selected by the first-time user. This is done by filtering our weakly correlated selections from the service processes in the form of associative rules.

As a result, we can obtain optimized preferences for the first-time user. The extracted preferences also apply to the experienced user so they can experience a better service. In the example given in this paper, a Thai restaurant's course menu was taken as an example. The result is the optimized menu course for a user that prefers $p_0$, $p_2$, $p_6$, $p_{10}$, $p_{14}$, and $p_{19}$ as shown in Table 2. The method is effective for applications such as travel planning services, learning courses, and interior design or fashion coordination services.

## 8. Conclusions

In this paper, we proposed a method to extract implicit preference based on process mining and data mining (associative mining). The model removes a refined preference model represented by Petri net and implicit preferences with a stronger correlation than the given explicit model. The proposed procedure was able to extract various associative rules and filter the rules to implicit output preferences based on Cook's distance and Pearson correlation coefficient. The process model supports associative mining by giving confidence in filtering up highly correlated rules and representing the combination of associative rules as a process model. The model serves as multiple options for items with multi-variate attributes. The proposed approach was evaluated and showed more than 70% of improvements even if the customer did not specify any interests towards any attributes for the item selection. The method is suitable for recommending a set of options rather than a single option. It is effective when used with services that offer many variations of combinations for the customer. For example, for the problem where more choices cause harder decision-making due to various possible combinations such as travel route planning services, meal ordering services, learning courses, and interior design or fashion coordination services. In future work, we will use the proposed method to identify sentiments in selecting options in a service. The proposed method is useful for supporting user experience by extracting customized preferences.

In future work, we will use the proposed method to identify sentiments in selecting options in a service. The proposed method is useful for supporting user experience by extracting customized preferences. We plan to extract the sentiments of previous customers to support the decision-making of new customers. Even though we can find the preference of *'no preference'* set for any attributes in the recommendation, the customer should be able to understand why the preference is recommended. We plan to provide sentiment recommendations for new customers so that the reason for the recommendation can be further understood and trusted. We will also consider correlation distances such as Mahalanobis distance and cosine similarity to compare the proposed method's effectiveness.

**Author Contributions:** Conceptualization, M.A.B.A. and S.Y.; methodology, M.A.B.A. and S.Y.; validation, M.A.B.A.; formal analysis, M.A.B.A.; investigation, M.A.B.A.; resources, M.A.B.A.; data curation, M.A.B.A.; writing—original draft preparation, M.A.B.A.; writing—review and editing, M.A.B.A., S.Y., A.K.M. and S.S.; visualization, M.A.B.A.; supervision, S.Y.; project administration, S.Y.; funding acquisition, S.Y. All authors have read and agreed to the published version of the manuscript.

**Funding:** This research was supported by Interface Corporation and Faculty of Engineering, Yamaguchi University, Japan.

**Data Availability Statement:** The dataset used in this study is partially and openly available in Kaggle. https://www.kaggle.com/datasets/anslab/thai-restaurant-dataset, accessed on 23 November 2022.

**Conflicts of Interest:** The authors declare no conflict of interest.

## Appendix A

Tables and figures used in the manuscript.

**Table A1.** Example for rules discovered by Apriori and averaged attribute values.

| Rules | Price | Rating | Calorie | Discount |
|---|---|---|---|---|
| $r_1$:('Appetizer2 = MiangKham', 90, 5, 280, 20) $\Rightarrow$ ('Dessert1 = CoconutCustard', 40, 4, 540, 10) | 65 | 4.5 | 410 | 15 |
| $r_2$:('Appetizer2 = GrilledShrimp', 83, 2.7, 125, 10) $\Rightarrow$ ('Dessert1 = CoconutCustard', 40, 4, 540, 10) | 61.5 | 3.35 | 332.5 | 10 |
| $r_3$:('Soup = TomYumKung', 90, 4.3, 229, 0) $\Rightarrow$ ('Dessert1 = CoconutCustard', 40, 4, 540, 10) | 65 | 4.15 | 384.5 | 5 |
| $r_4$:('MainDish2 = ChickenPorridge', 50, 4.2, 228, 25) $\Rightarrow$ 'Dessert1 = CoconutCustard', 40, 4, 540, 10) | 45 | 4.1 | 384 | 17.5 |
| $r_5$:('MainDish2 = KanomJeenNamYa', 45, 2, 81, 5) $\Rightarrow$ ('Dessert1 = MangoStickyRice', 87, 5, 270, 0) | 66 | 3.5 | 175.5 | 2.5 |
| $r_6$:('Dessert1 = CoconutCustard', 40, 4, 540, 10) $\Rightarrow$ ('Appetizer2 = MiangKham', 90, 5, 280, 20) | 65 | 4.5 | 410 | 15 |

**Table A2.** Pattern of preference and its correlation $PR, PC, PD, RC, RD, CD$ between attributes.

| Pref. | P | R | C | D | Rules | PR | PC | PD | RC | RD | CD |
|-------|----|----|------|----|-------|------|-------|-------|-------|-------|------|
| $p_0$ | **NI** | **NI** | **NI** | **NI** | **52** | **0.53** | **−0.1** | **−0.35** | **−0.06** | **−0.05** | **0.13** |
| $p_2$ | **NI** | **NI** | **Low** | **NI** | **46** | **0.54** | **0.22** | **−0.33** | **0.07** | **−0.03** | **0.15** |
| $p_4$ | NI | NI | High | NI | 6 | 0.88 | −0.9 | −0.99 | −0.58 | −0.8 | 0.95 |
| $p_6$ | **NI** | **High** | **Low** | **NI** | **42** | **0.45** | **−0.23** | **−0.35** | **0.12** | **−0.06** | **0.16** |
| $p_8$ | NI | High | High | NI | 6 | 0.88 | −0.9 | −0.99 | −0.58 | −0.8 | 0.95 |
| $p_{10}$ | **Low** | **NI** | **Low** | **NI** | **40** | **0.54** | **−0.34** | **−0.3** | **0.13** | **−0.02** | **0.36** |
| $p_{12}$ | Low | NI | High | NI | 6 | 0.88 | −0.9 | −0.99 | −0.58 | −0.8 | 0.95 |
| $p_{14}$ | **Low** | **High** | **NI** | **NI** | **42** | **0.41** | **−0.06** | **−0.37** | **−0.09** | **−0.09** | **0.2** |
| $p_{17}$ | Low | High | High | NI | 6 | 0.88 | −0.9 | −0.99 | −0.58 | −0.8 | 0.95 |
| $p_{19}$ | **Low** | **High** | **Low** | **NI** | **36** | **0.45** | **−0.36** | **−0.33** | **0.2** | **−0.07** | **0.38** |

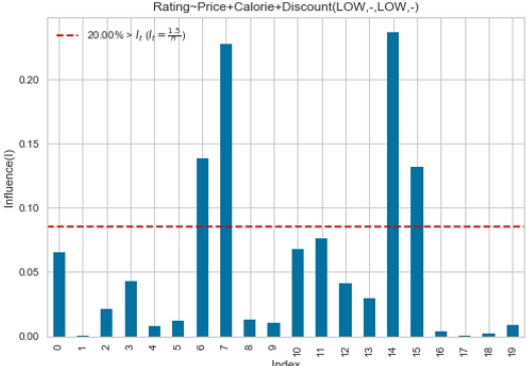

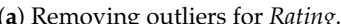

(**a**) Removing outliers for *Rating*.

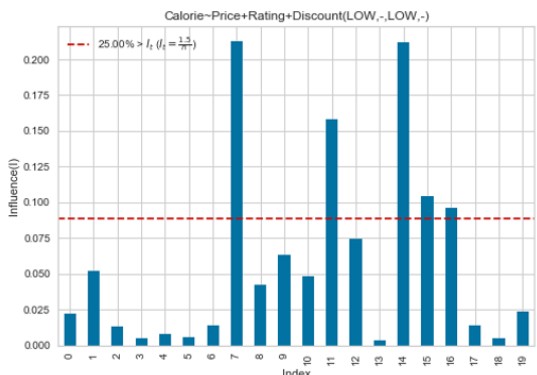

(**b**) Removing outliers for *Calorie*.

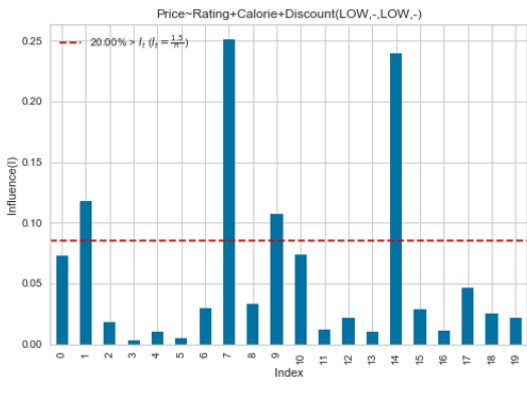

(**c**) Removing outliers for *Price*.

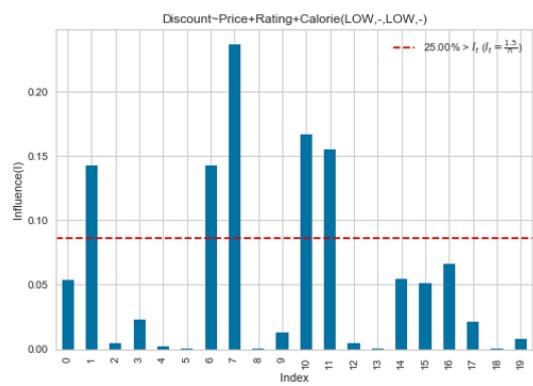

(**d**) Removing outliers for *Discount*.

**Figure A1.** Outlier detection based on Cook's distance. The red lines represent the cut-off threshold.

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
