# Peer review of "Refining Preference-Based Recommendation with Associative Rules and Process Mining Using Correlation Distance"

_2504-2289, doi:10.3390/bdcc7010034_

Round 1

Reviewer 1 Report

The paper is well structured. The introduction is clear and in line with what is reported in the conclusions. The two figures 2 and 7 are too crowded with the related writings and are difficult to read. I suggest a better rearrangement of the notes so that you can read the figures better. For the rest, I am in favor of publication.

Author Response

Thank you for your indications. Please see the reply letter.

Reviewer 2 Report

The paper proposes a new method for the analysis of customers’ preferences. The authors are using Process mining and associative mining techniques in this sense. The Process Mining algorithm chosen by the authors is Inductive Miner (from ProM Framework) whose output is a sound Petri Net. In order to extract the association rules, low confidence, and lift value, the authors are using the Apriori algorithm.

For modeling the course menu of a Thai restaurant, a PN is proposed to be used. The menu has 8 categories and they are characterized by 4 attributes: price, rating, calorie, and discount. First, the association rules are extracted, followed by the calculation of the correlation between attributes. High-distance values are removed. In this way, the optimized preferences of the customers are obtained. The approach is validated by using a sales log consisting of 4 items and at least 60000 data rows. The Apriori algorithm extracted 40 associative rules.

The flow of the paper is easy to understand. The introduction clearly presents the main contributions of the paper and defines the research area. A section presenting the preliminary concepts used in the paper also exists (concepts like Process Mining, Association rules, and Cook’s distance are introduced). The related work section visibly delimitates the boundaries of the research by describing similar approaches proposed by other researchers. The method proposed by the authors is well presented and the validation is made by using a case study of a Thai restaurant (but can be extended to other domains). Conclusions reflect the results got by the authors.

Minor modifications:

Some silent transitions are used with no sense (see figure 7a)). An improvement of the algorithm can be made. Indeed, IM uses silent transitions, but when constructs like parallelization are needed.

Page 6/17 -> Figure 2. The model for the item selected in the course menu (original model).

Page 15/17 -> move the figure  somewhere above

Author Response

Thank you for your indications. Please see the attached reply letter.

Reviewer 3 Report

The manuscript is well-written and presented appropriately. I have the following suggestions for authors to improve their work.

--- I have found some minor grammatical mistakes.  Authors are requested to give proofread before the final submission. 

---The research gap is a little unclear; please elaborate on it; what is the motivation behind carrying on this study? What is the ultimate benefit? 

--- Procedure 01 and some figures Can be attached as supplementary material. I suggest authors remove less important figures from the main manuscript.

--- I suggest authors provide a comparative study of the most relevant published work on the topic. Authors can present it in a table. 

Author Response

(The authors gave the same response as above.)

Reviewer 4 Report

In this research, authors have proposed a method for extracting associative rules with a high degree of correlation between multivariate characteristics based on intuitive preference settings, process mining, and correlation distance.

I). What are the motivations of your work.

II). Please add some more relevant literature to the introduction section. In Table 1, the provided references for clustering analysis are insufficient.

III). Why the authors are interested in Cook’s distance measure. Can we replace it by Euclidean distance. Also, please confirm whether the Cook’s measure satisfies the axiomatic properties of metric, particularly triangular inequality axiom?

IV). In Figure 2, it would be preferable to relocate the text to the outside of the diagram.

V). Describe the time complexity of the suggested algorithm in detail.  

VI). Mathematical contribution of the work is very limited. Also the presentation of the paper needs improvement.

Author Response

(The authors gave the same response as above.)

Reviewer 5 Report

Studies associated with user preferences have grown in size and depth because of the assistance of the internet and novel ideas concerning customer preferences have been developed. The work being reviewed attempted to contribute to the discussion in the area exploring associative rules, process mining and correlation distance. The authors use data to argue their case. In this regard, the work potentially contributes to the literature in the area. However, to heighten its robustness, the manuscript should be revised to address the following issues:

-The authors should discuss future research aspects of the work

- Add some more 2022 articles, possibly about five.

Author Response

(The authors gave the same response as above.)

Round 2

Reviewer 4 Report

Comment 3, responce is unstaisfactory.

Author Response

Thank you for your indications.  Please refer to the attached file for the answers.
